cognition/psychology

human communication, synchrony, triads, laughter

**Author for correspondence:**
Rick Dale
e-mail: rdale@ucla.edu

# Body synchrony in triadic interaction

Rick Dale[1,3], Gregory A. Bryant[1,3], Joseph H. Manson[2,3] and Matthew M. Gervais[4,5]

[1]Department of Communication, [2]Department of Anthropology, and [3]Center for Behavior, Evolution, and Culture, University of California, Los Angeles, CA, USA
[4]Centre for Culture and Evolution, and [5]Department of Life Sciences, Brunel University London, Uxbridge, UK

 RD, 0000-0001-7865-474X; GAB, 0000-0002-7240-4026

Humans subtly synchronize body movement during face-to-face conversation. In this context, bodily synchrony has been linked to affiliation and social bonding, task success and comprehension, and potential conflict. Almost all studies of conversational synchrony involve dyads, and relatively less is known about the structure of synchrony in groups larger than two. We conducted an optic flow analysis of body movement in triads engaged in face-to-face conversation, and explored a common measure of synchrony: time-aligned bodily covariation. We correlated this measure of synchrony with a diverse set of covariates related to the outcome of interactions. Triads showed higher maximum cross-correlation relative to a surrogate baseline, and 'meta-synchrony', in that composite dyads in a triad tended to show correlated structure. A windowed analysis also revealed that synchrony varies widely across an interaction. As in prior studies, average synchrony was low but statistically reliable in just a few minutes of interaction. In an exploratory analysis, we investigated the potential function of body synchrony by predicting it from various covariates, such as linguistic style matching, liking, laughter and cooperative play in a behavioural economic game. Exploratory results do not reveal a clear function for synchrony, though colaughter within triads was associated with greater body synchrony, and is consistent with an earlier analysis showing a positive connection between colaughter and cooperation. We end by discussing the importance of expanding and codifying analyses of synchrony and assessing its function.

## 1. Introduction

Overt behavioural coordination among members of a species is a common phenomenon, supporting conflict, mating and mutual survival. Among non-humans, examples abound, from single-celled organisms to our closest relatives. *Dictyostelium discoideum*

(slime mould) transitions from relatively independent single cells into an organized multicellular collective when resources become scarce [1]. Mosquitoes may synchronize their wing flaps during mating [2], fish schools emerge from local interactive dynamics [3], and common chimpanzees appear to coordinate hunting in groups [4]. Humans show pervasive behavioural coordination. Indeed, people exhibit a large number of collective behaviours, spanning group sizes and across different forms of behaviour [5].

In human face-to-face conversation, various types of behaviour have been investigated. Many distinct behaviours may reveal correlated structure among members of an interaction [6–8]. One major theme in research on human coordination is *behavioural synchronization*. As a general technical term, synchronization describes the tendency for behavioural patterns to become more similar while two or more people interact. In this paper, we focus on a behaviour that has been widely studied: bodily movement during face-to-face conversation. Though overt body movements can serve as a volitional signaling device, such as assertiveness or aggression, they may also provide more subtle, implicit signals of interest or engagement. Numerous studies have offered evidence for synchrony between interaction partners in the subtle fluctuations of their body movement. Relative phase analysis has shown that in casual interaction, body rhythms tend to be similar [9]. Cross-correlation has also been used to show a low but reliable covariation in overall body movement during conversation [10,11]. The general finding that body movement shows patterns of synchrony between dyads has been shown in several studies (e.g. [12–21]). The same tendency towards interactive synchrony has also been shown in other types of human behaviour such as eye movements [22], expressive emotion [23], speech-related convergence [24–28], and more. All of these behaviours may contribute to a multidimensional dynamic coordination between two people who are interacting using language [7].

Despite the many demonstrations of synchrony, its functional significance remains unclear. In some cases, synchrony may indicate a desire to bridge a social gap when there is a perceived breakdown or potential breakdown in interaction [29–31]. This would suggest that body synchrony appears during disrupted or unstable affiliation. Other studies have found the opposite. When affiliation is high, cross-correlation structure between members of a dyad appears to be higher [10], and when two people's eye movements are in coordination, it suggests increased understanding [22,32,33]. In clinical contexts, patient–therapist alliance may be indexed by synchronized non-verbal behaviours [17].

In general, it is clear that temporal relationships among behaviours during interaction can highlight an array of task goals and contexts. However, synchrony cannot account for all aspects of human interpersonal dynamics. 'Pure synchrony' would quickly lead to dysfunction in human interaction, and some studies have shown subtler relationships across different levels of analysis (e.g. linguistic: [34]). If humans purely synchronized, complex tasks could not be performed, as members of a pair or group have to mix behavioural strategies to make possible varied aims of conversations and other social tasks [35]. These intuitive considerations suggest that synchrony cannot be the sole structural ingredient of interactions, and some recent empirical studies on different interactive tasks also suggest it cannot be the only such ingredient (e.g. [36–38]). Indeed, though Louwerse *et al.* [7] refer to their findings as 'synchrony', they use the term to refer to a number of different coordination patterns emerging in their expansive multimodal analysis.

Many of the studies cited above are task-oriented. Participants work together to solve a particular puzzle or communicative goal. Such tasks are no doubt critical in our understanding of the temporal coordination of human interactions. It is equally important to determine the presence and role of synchrony in natural face-to-face interaction. It is reasonable to identify what is sometimes termed 'phatic communion'— casual face-to-face interaction—as one of our species' longest-standing forms of interaction. Prior work suggests that there is a relationship to affiliation in such interactions (e.g. [10,24,25]), but the mechanism underlying this effect is still unclear. How this effect works, and how it relates to bodily synchrony, is still not well understood.

Here we analyse a corpus of triadic human interaction. Triads offer an opportunity to tap into the potential mechanisms underlying a variety of discourse processes, such as turn-taking, reference and conceptual pacts [39–42]. This prior research motivates the present focus on triads as a source of information about body synchrony itself. Do triads show group synchrony, reflecting a general *average affiliation* among its three members? Or do triads show distinct patterns of synchrony in their component dyads, potentially reflecting mutual affiliation only among those pairs? Does increased bodily synchrony between a pair predict that they will form an alliance in future interactions? Using cross-correlation in the signature of their body movements, we show that on average they reveal behavioural synchrony in the component dyads. The triads also show synchrony between the *pairs* of conversants, suggesting that the triad is loosely 'moving together' during group interaction. In order

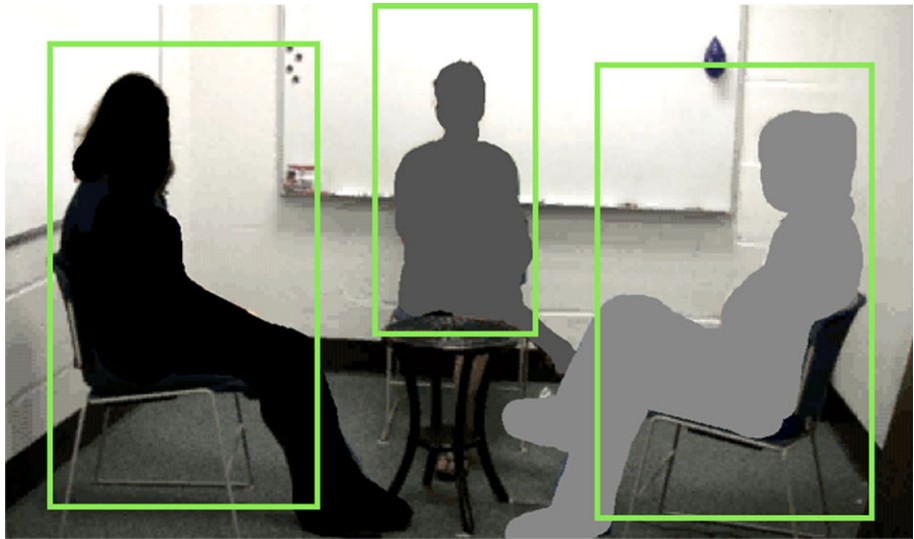

**Figure 1.** An example frame of a conversational video with left, centre, and right members. The green boxes reflect regions of interest. The optic flow method of Barbosa *et al.* [44] extracts the pixel changes and measures magnitude of change that can serve as a proxy measure to body movement of each member (see also [10]).

to determine the potential functional role of this synchrony, we predict it from a variety of individual differences and outcome measures in an exploratory regression analysis.

# 2. Methods

A full description of the participants and the tasks can be found in Gervais *et al.* [43]. This includes detailed summary of the interactive instructions, gameplay and procedures, individual differences and questionnaires, and so on. In sum, participants (described below) arrived to a laboratory to engage in 'small talk with strangers'. Triads were then seated in a room and video-recorded having a 10 min conversation about any topic of their choosing. After 10 min, each participant was seated in a private cubicle in front of a computer where they played a simultaneous one-shot Prisoner's Dilemma (PD) game with each of the other two individuals from their conversation. Each participant had a fund of $3, and they were told they could 'transfer' or 'keep' it, that the other players had the same choice, and that transferred money would be doubled, so that if they transferred $3, the recipient would receive $6. Many other variables were measured, with an overall empirical goal of assessing how subclinical psychopathy contributes to conversational behaviour and cooperative interaction. Here we focus on particular features of the dataset that guide the current analysis.

## 2.1. Participants

The original study included 105 undergraduates at the University of California, Los Angeles (UCLA). All participants were given $10 to show up to the task, in addition to receiving course credit (approx. 90% of participants were also seeking course credit). They were all native English speakers, and represented a range of ethnic backgrounds from UCLA's diverse undergraduate population. Groups included entirely male (15) and entirely female (20) triads. Members of the triads did not know each other before their participation.

## 2.2. Time series from conversational video

Triads sat around a small table, facing each other. They were asked to converse about any topic they wished, and were recorded for 10 min using a Canon Vixia HV30 camcorder, combined with an Audio-Technica U841A omnidirectional condenser microphone.

We measured the body movement of each member of the triad. An optic flow method calculates how pixels are changing from frame to frame to determine the magnitude of movement in the video [44,45]. We isolated regions for each triad member, shown in figure 1. These regions were delimited in

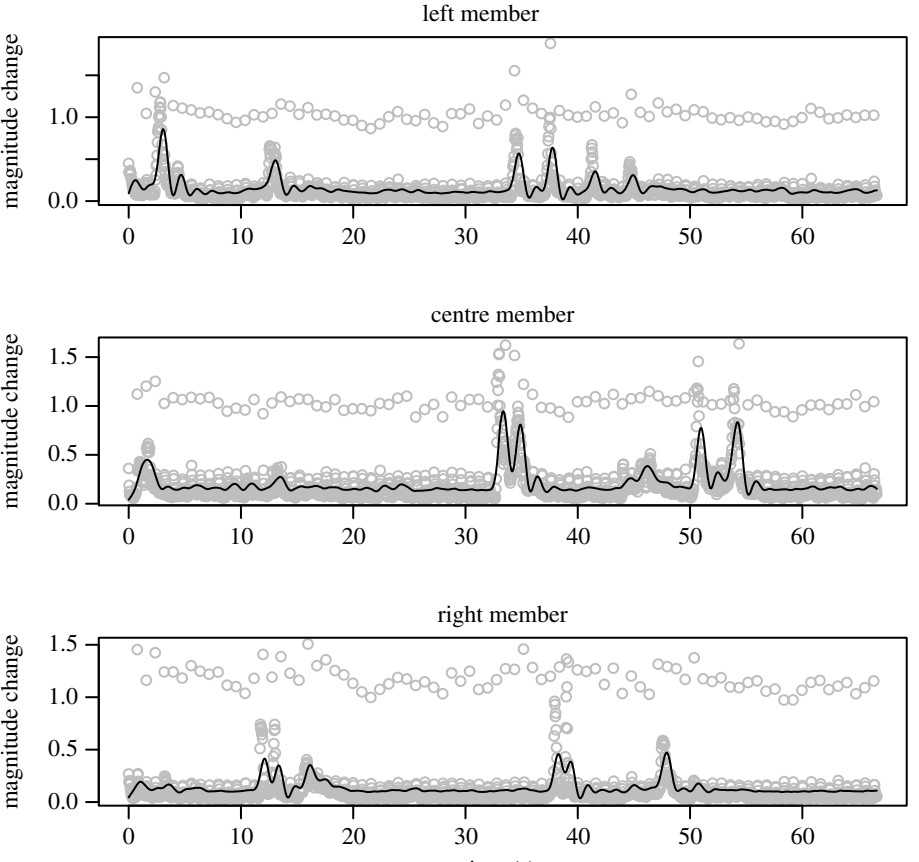

**Figure 2.** Raw magnitude (grey circles) from the optic flow method includes noise due to video compression and lighting in the conversational video. Using a low-pass filter (black lines) we obtain a proxy measure of body motion over time that omits these high-frequency artefacts.

such a way as to maximize coverage of the whole body. This mostly included upper legs, torso, and head, because the lower legs might move in such a way as to overlap with or enter the area of another interlocutor's region. We scanned entire videos to ensure that a region chosen did not ever include part of the body of one of the other triad members. Changes in pixels in each region will reflect likely body movement of the corresponding participant. This measure provides a time series for each member, at the frame rate of the video. An example trio of time series is shown in figure 2, grey dots.

Time series were filtered to avoid high-frequency noise in influencing body correlation analysis. Such noise can derive from at least two main sources: high-frequency fluctuations in the video (such as from fluorescent lighting) and from the video compression algorithm (see [10] for a discussion of filtering and validation of frame-differencing with source body movement). All videos were in AVI, with approximate data rate of 3500 kb s$^{-1}$, size $960 \times 540$ and 30 fps. We subjected the extracted optic flow signals to a low-pass eighth-order Butterworth filter in R (creating a sharp dB cut-off) for a relatively low frequency (0.05 of the Nyquist frequency). In order to find this cut-off, we were guided by prior video differencing research that obtained body motion signals from similar video (e.g. second-order filter at 0.2 Hz cut-off: [10,11]). Because the quality of the present videos differed, and there appeared to be more periodic noise, we used a more aggressive set of parameters for this filter, and judged the fit to a sample of the raw movement data. The same parameters were used across all videos and these parameters were chosen before analysing the main results. As a stop-pass filter, the cut-off frequency under these conditions occurs in the range of [0.6 Hz, 1.0 Hz]. Though this is lower than the cut-off of prior work, visual inspection of the filtered time series confirmed that this improved approximation of the body movement during the interaction, shown also in figure 2, solid black lines. Prior to analysis, to ensure that the first few seconds of the filtered time series did not drive our results, we trimmed 200 samples (approx. 6 s) from the start of the time series consistently for all triads.

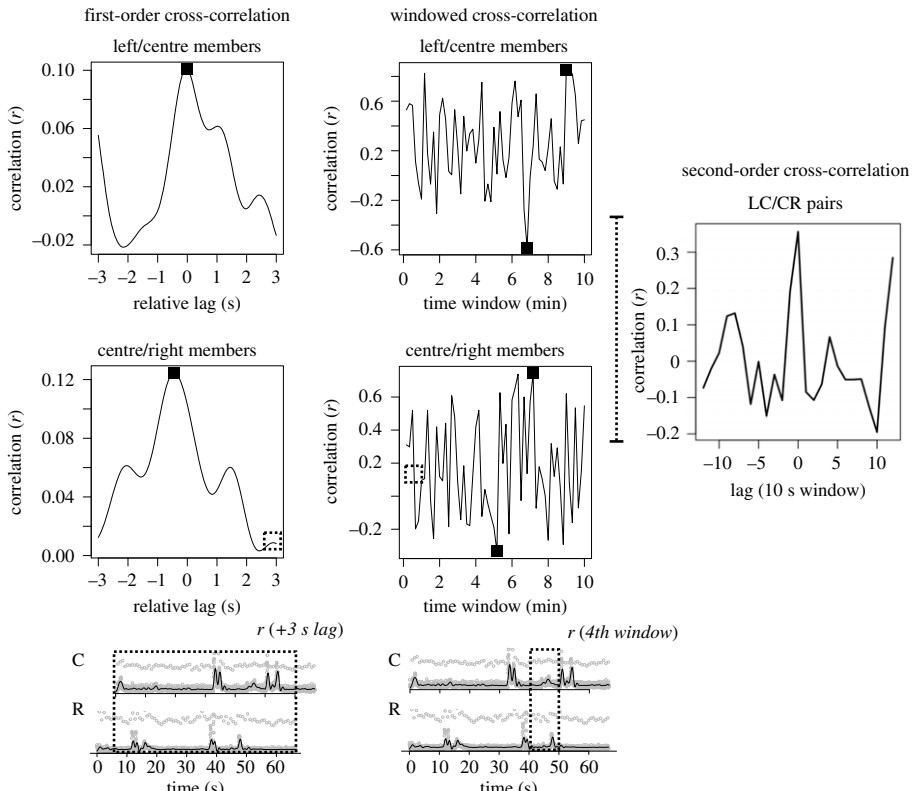

**Figure 3.** On the left side is an aggregate reflection of correlation between two people, the cross-correlation function (triad #11). At lag of 0, if two people are on average correlated in their movement, then we will see maximum correlation. When it is shifted (illustrated below, with 3 s), this maximum should drop off as we lag the time series relative to each other. Black dots represent the dependent measures extracted for each pair: maximum correlation, minimum correlation, cross-correlation function maximum and its corresponding lag. On the right side, we show the ebb and flow of body correlation between two members of an example triad (#11). In windows of 10 s, the correlation can approach maximum ($r = 1$), and occasionally members of a triad show negative body correlation ($r < 0$). Next to that, the second-order measure of triadic synchrony was based on the cross-correlation function of these two time series. This assesses whether a triad's component dyads covary in synchrony.

# 3. Measures

## 3.1. Dependent variables: body movement

We tested perhaps the most common measure of synchrony used in interaction research, namely measurements based on *time-aligned bodily covariation*. These are most often measured via cross-correlation as a signature of the coordination taking place during the triadic interaction. Though there are numerous measures available [46] and debate about use of correlation in dyadic time-series contexts [47], cross-correlation is often used because, compared with alternatives, its computational basis is relatively transparent, and interpretation of its output is relatively clear. Importantly, we compare observed cross-correlations in multiple ways to a surrogate statistical baseline. By combining the most common measure of synchrony to our diverse set of covariates, our study thus serves as a test of the potential role synchrony may be playing.

An example is shown in figure 3, using both windowed correlation and a cross-correlation function for two pairs of one triad (left–centre, centre–right pairs). The windowed correlation scans 10 s segments of the motion time series (cf. [48]). The 10 s duration was chosen because it should effectively bound interpersonal bodily correlation that is in synchrony with a decay of several seconds (e.g. 3–4 s: [11]), while being fine-grained enough to observe the variation in this correlation across the interaction. Windows were extracted at each 10 s interval and did not overlap (giving approx. 60 windows per triad). In each 10 s segment, we calculated the correlation in the body movement of each pair composing the triad. A general observation here is that synchrony varies rather widely across an interaction, suggesting interaction partners form a more loosely coupled system than may be assumed from average analysis (cf. [49,50]).

Across this windowed calculation of correlation, we extracted the *maximum correlation* and *minimum correlation* observed in each pair. We also calculated an aggregate cross-correlation function of these pairs, shown in figure 3, left. This function specifies the temporal relationship between body movements of the pairs. If a pair tended to move at almost the same time, then correlation will be highest at a lag of 0, where the two time series overlap. As lag is changed, we overlap or 'slide' the time series over each other, shifted in accordance with each lag. If two people are coordinating, increasing lag will reduce the correlation in their body movement. We extracted the *maximum cross-correlation value* as the maximum of this function, and we also took the *maximum correlation lag*, indicating the lag at which that maximum occurred. If two people were moving synchronously, this maximum correlation lag would occur at 0.

To assess whether there is *triadic* synchrony, we use windowed cross-correlation as a basis for what can be termed a *second-order* correlation. Taking the window-wise correlations shown in figure 3, middle, we compute the cross-correlation of these new time series. We take the windowed cross-correlation of all three pairs (left–centre, left–right, etc.) and calculate a cross-correlation. This reflects the tendency for the dyads to covary *together*. If the component dyads are correlated, then this second-order cross-correlation should also show a peak at a lag of 0 relative to a surrogate baseline.

All analysis scripts and raw data can be found at https://github.com/racdale/triadic-bodily-synchrony.

## 3.2. Baseline variables: surrogate pairs

To test whether the observed correlation differs from a baseline, we constructed surrogate pairs: pairings of one participant with participants in other triads. For each triad, we constructed a surrogate condition using data from *all* other triads, substituting one of these other participants (e.g. left) into exactly the same analysis. Because these time series are drawn from different triadic interactions, their cross-correlation ought to be closer to 0 than in the observed data.

## 3.3. Additional measures

An array of measures was collected in the original study in Gervais *et al*. [43]. Some measures characterized the individuals in the interaction (e.g. median household income, sex, psychopathy and more [43]). Other measures reflected the quality of the interaction, its potential outcome, and participant similarities. Common ground was coded from the conversation video, as whether or not a dyad discovered one of a set of commonalities (yes or no; same academic major, an acquaintance in common, etc.). Cultural style similarity was coded by preparing a set of still image stimuli in which all exposed skin was replaced by a colourless mask (leaving only clothing, hair and jewelry), and raters then scored pairs of these images by how similar they were culturally. Other variables included similarity in ethnicity, language style matching and cooperation versus defection in a one-shot PD game. All of these measures were taken after the face-to-face interaction.[1] The measures are therefore a potential window onto the role that bodily synchrony might play in generating affiliation or alliance in subsequent ratings.

Importantly, as just noted, in the original task in Gervais *et al*. [43], most variables did not have temporal priority such that they could be interpreted as potential causal variables of bodily synchrony. For example, the PD variable was collected *after* interactions took place, and so the PD outcome could not be interpreted as a *cause* of bodily synchrony. Here we simply take these variables as correlates that may speak to functional considerations, without making strong commitments on the causal basis for their contribution. All measures are summarized in table 1.

# 4. Analysis and predictions

## 4.1. Dyads show body correlation in time

If dyads synchronized, we should expect that component dyads would show increased body coordination relative to the surrogate baseline. In this case, maximum synchrony across the interaction, maximum cross-correlation and the relative distribution of the maximum lag location should all indicate synchrony.

---

[1]While measured after the interaction, in some cases the measures are proxies of traits observable by participants at the start of the interaction, which could have influenced synchrony (such as ethnicity and style similarity).

**Table 1.** Measures used in the body analysis and additional measures.

| variable | min | *M* | max | s.d. |
|---|---|---|---|---|
| *body-dependent variables* | | | | |
| maximum correlation (*r*) | 0.41 | 0.81 | 0.97 | 0.10 |
| minimum correlation (*r*) | −0.73 | −0.48 | −0.32 | 0.10 |
| cross-correlation at lag 0 (*r*) | −0.11 | 0.05 | 0.39 | 0.09 |
| triadic correlation (*r* at lag 0) | −0.19 | 0.16 | 0.51 | 0.15 |
| *additional measures* | | | | |
| sex (1 = female) | 0.00 | 0.57 | 1.00 | 0.49 |
| childhood income (*z*) | −1.72 | 0.02 | 2.20 | 1.00 |
| psychopathy (*z*) | −1.89 | 0.00 | 3.23 | 1.13 |
| attractiveness (*z*) | −1.78 | −0.03 | 2.31 | 1.02 |
| cultural style match (*z*) | −2.17 | 0.00 | 2.27 | 1.00 |
| language style match (LSM) | 0.52 | 0.82 | 0.95 | 0.08 |
| perceived common ground (0/1) | 0.00 | 0.44 | 1.00 | 0.50 |
| conversational interruptions (rate) | 0.00 | 0.40 | 2.65 | 0.46 |
| Prisoner's Dilemma game (1 = cooperate; 0 = defect) | 0.00 | 0.63 | 1.00 | 0.48 |
| perceived warmth (*z*) | −3.11 | −0.07 | 1.27 | 1.01 |
| perceived competence (*z*) | −0.65 | 0.02 | 1.16 | 0.99 |
| total laughter (count) | 5.00 | 29.11 | 55.00 | 13.01 |
| colaughter (%) | 13.00 | 41.43 | 75.00 | 16.87 |

## 4.2. Triads show correlation in time

If triads synchronized together, then the correlations between component pairs ought to be correlated with each other over time. In other words, if the participants seated on the left and middle were correlated during a given time frame, we might expect the same correlation to be occurring with the person on the right. This can be quantified relative to a surrogate baseline by calculating a second-order cross-correlation: The sliding 10 s windowed correlation values should themselves correlate from pair to pair. We tested this by calculating this second-order cross-correlation function and compared it with the same value measured using the surrogate baseline.

# 5. Exploratory regression with additional measures

What communicative or affiliative function would this general body correlation serve? Because the conversation took place before the collection of other measures, any relationship among these variables can be interpreted as a *potential* role for bodily synchrony in driving interpersonal outcomes. We recognize, of course, that this is a correlational study from a pre-existing corpus, so any causal verbiage should be used with care. The temporal relationship in the original data collection procedures permits us to use body motion as a dependent variable, predicted by the array of other measures. Any relationship between the additional measures and the body correlation could be interpreted as a *potential* outcome of bodily synchrony. We take a repeated-measures approach to our design, to increase within-triad power and increase the probability of identifying any potential underlying patterns. By taking the 10 s windows and treating these as the repeated measurement, we can use the fixed predictors (PD outcome, common ground, cultural style matching, etc.) as repeat covariates.

To build this exploratory regression model, we aggregate the mean values across the array of measures in table 1, for each dyad within the triad. For example, the values of the individual differences and interactive variables from table 1 would be averaged for left–right members of the interaction. These would then be paired with the outcome variable of how much body correlation they showed in *each* 10 s window.

**Table 2.** Tests of general behavioural correlation relative to baseline. Model form: $dv \sim surrog + (1 + surrog \mid triad)$. Note: results here are from a mixed-effects model, and maximal random-effect structure did not always converge. Results were thus checked with analogous paired-samples $t$-test (observed–surrogate), with results significant and consistent.

| dependent measure ($r$) | observed | surrogate | $t$ | $p$ |
| --- | --- | --- | --- | --- |
| maximum correlation | 0.81 | 0.74 | 6.78 | $1.2 \times 10^{-11}$ |
| minimum correlation | −0.48 | −0.55 | 6.26 | $4.0 \times 10^{-10}$ |
| cross-correlation at lag 0 | 0.05 | 0.00 | 4.02 | $5.8 \times 10^{-5}$ |
| triadic correlation | 0.16 | 0.10 | 3.50 | 0.00047 |

In order to control for within-triad unique patterns that may generate our results, we take a mixed-effects regression approach. Using lmer in R, we built a mixed-effects regression model and specified an intercept term for each triad. Though it is desirable to maximize this random effect structure (nested slopes; [51]), the number of additional variables was simply too large for the model to converge. In addition, choosing which variables to omit or include should be guided by theoretical concerns, and we explicitly take an exploratory stance here. For this reason, we adopted an admittedly liberal approach in our analysis, embracing the exploratory nature of the analyses. We interpret any significant coefficients as a *potential* functional relationship between body correlation and interactive and individual measures; such significant coefficients could be the grist for future experiments or analyses.

Based on the many prior findings summarized in the Introduction, we may render a few predicted outcomes. First, body correlation may signal affiliation, so that perceived warmth and common ground may positively relate to it. If co-movement reflects the emergence of alliance during the interaction, then we could predict that cooperation in the PD game may occur more frequently. By factoring in a variety of variables in an exploratory regression model, we can test the relative contribution of these additional measures.

# 6. Results

## 6.1. Body synchrony within triads

In all results reported in this section, unless otherwise noted, linear mixed effects models were used with the maximal random effects structure [51]. The primary fixed factor was a variable that specified observed data ($N = 35$ triads × 3 pairs = 105) versus surrogate baseline ($N = 35$ triads × 3 pairs × 34 surrogates = 3570). Nested slopes were included by specifying a random structure at the triad level. For all models reported in table 2, we ensured variables and residuals were approximately normally distributed, and also confirmed our results with a corresponding paired-samples $t$-test that directly contrasted observed statistics with a mean surrogate. These yielded the same patterns of significance.

Maximum correlation shown in 10 s windows was higher in observed data than in the surrogate condition, $M = 0.81$ versus 0.74, $t = 6.78$, $p < 0.0001$. In addition, minimal correlation in 10 s windows was also significantly higher in the observed data, $M = -0.48$ versus −0.55, $t = 6.26$, $p < 0.0001$. This negative correlation in observed dyads was reliable. Triads showed significantly *negative* minimum correlation in their 10 s windows, indicating that during a casual interaction there were statistically significant periods of time during which members were negatively correlated, one-sample $t$ at triad level: $t_{34} = -43.83$, $p < 0.0001$. These results are summarized in table 2.

On average, across interactions, participants showed positive bodily synchrony. Figure 4, left, shows the average cross-correlation function across all pairs of a triad. In general, we see a low but statistically reliable peak at lag of 0. Statistical significance of this peak was assessed with a linear mixed effects model, comparing observed data with the same cross-correlation function at lag 0 for the surrogate condition (shown in grey in figure 4, left). This cross-correlation is indeed quite low, though it is comparable to prior tests for average body correlation across a conversation [10,11]. Table 2 shows the output of the linear mixed effects model at lag of 0, showing that despite a small effect, it is reliable when compared with the baseline surrogate profiles at the same lag. When data were separated by pairs (e.g. left–middle chairs), we found the same statistically significant effect at lag of 0, so the overall effects were not due to a particular position of participants in the video frame. When we

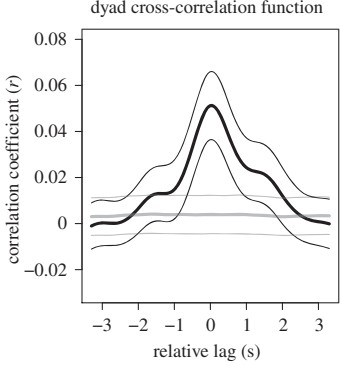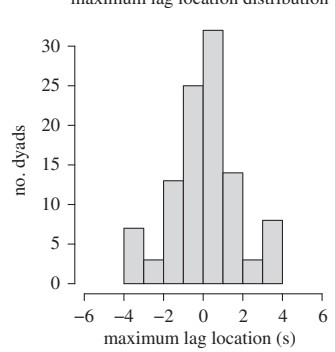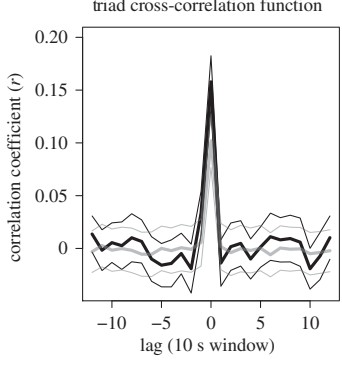

**Figure 4.** On the left, the average cross-correlation function for observed data (black), with narrower lines reflecting standard error (using conservative d.f. $= N_{triads} - 1 = 34$). At its peak, this function shows a significantly greater value than the surrogate pairs, shown in grey. In the middle, the distribution, across dyads, of the lag where the maximum cross-correlation is observed ($N = 35$ triads $\times$ 3 pairs = 105 dyadic observations). On the right, the cross-correlation function of the triads, based on how correlation among the pairs changes over time. In a narrower effect near the lag of 0, triads show higher relative correlation.

plotted a histogram using our lag measure—the lag location of the maximal cross-correlation function—we obtained a distribution that was centred near 0, with the large majority of dyads showing their maximum cross-correlation at or near synchrony (figure 4, middle).

Are triads in synchrony together? We used a second-order correlation to test whether body movement covaried across all pairs of the triad. In other words, does the extent to which left–middle members were correlated relate to how middle–right members were correlated, and so on? By computing the cross-correlation function of the 10 s window $r$ values shown in figure 3, we see a reliable lag-0 correlation (as above, observed pairs $N = 105$ and surrogate condition $N = 3570$). Figure 4, right shows that triads tended to be correlated in a manner similar to their component pairs, and perhaps at a higher degree, approximately $r = 0.16$. This is significantly greater than the correlation observed across surrogate pairings. Our surrogate baselines in this analysis are a conservative test of this triad-level synchrony, because each surrogate cross-correlation is computed using one actual participant of a triad. We would expect, by chance, some cross-correlation within the surrogate triad because the extent to which a baseline in (say) the left–middle surrogate is correlated, should correspond with the middle–right surrogate, since one of the surrogate members (the observed pair) correlates with each other (figure 3, right). Nevertheless, the observed triad second-order correlation was significantly higher, $r = 0.16$ versus 0.10, $t = 3.50$, $p < 0.0005$.

Results here demonstrate that triads were in synchrony. First, pairs showed a low and statistically reliable body correlation relative to baseline, at about $r = 0.05$. Though this seems low, this correlation can be quite high in some 10 s periods of time (and even, though rarely, near maximum correlation of 1.0). In addition, triads showed second-order correlation that exceeded baseline: triads were moving together *together*, at $r = 0.16$. As in prior work, we show that humans are in a low 'hum' of behavioural synchrony across minutes of interaction.

## 6.2. Exploratory regression model

The exploratory regression model revealed a relationship with only three of our additional variables. To assess potential relationships, we used a conventional significance cut-off ($p < 0.05$), though the results below would accommodate correction. Nevertheless, any results here should be interpreted cautiously, as it is a purely exploratory analysis. The regression model was again a mixed-effects model predicting the $N = 6228$ windowed correlations (across 105 pairs in 35 dyads) using predictors in the form of several outcome and individual covariates.

Among the coefficients that appear to reveal some potential signal, two of these relationships are paradoxically *negative*. Participants who were judged by raters as having *lower* levels of cultural style similarity tended to show *greater* body synchrony. A second variable also exhibited a negative relationship with body motion: Participants who show increased body correlation tended to have *lower* language style matching. This coefficient has a magnitude suggesting a stronger relationship than the one with cultural style matching. Table 3 shows unstandardized coefficients ($B$) and their standard errors (s.e.). Laughter is also related to body synchrony. Triads who had more colaughter

**Table 3.** Regression model predicting correlation in 10 s segments. Model form: $r_w \sim$ sex $+ \ldots +$ laughs $+$ (1 | triad). Bold text indicates significant predictors at the $p < 0.05$ level (see main text for summary of exploratory approach).

| additional measure | B | s.e. | t | p |
|---|---|---|---|---|
| sex (male or female) | −0.006 | 0.017 | −0.35 | 0.727 |
| childhood income (z) | 0.015 | 0.009 | 1.64 | 0.101 |
| psychopathy (z) | −0.007 | 0.009 | −0.81 | 0.418 |
| attractiveness (z) | −0.016 | 0.009 | −1.82 | 0.069 |
| **cultural style match (z)** | **−0.014** | **0.005** | **−2.60** | **0.009** |
| **language style match (LSM)** | **−0.266** | **0.081** | **−3.29** | **0.001** |
| perceived common ground (0/1) | −0.010 | 0.012 | −0.81 | 0.420 |
| conversational interruptions (rate) | 0.019 | 0.018 | 1.11 | 0.269 |
| Prisoner's Dilemma game (1 = cooperate; 0 = defect) | −0.021 | 0.017 | −1.25 | 0.212 |
| perceived warmth (z) | −0.009 | 0.009 | −1.03 | 0.302 |
| perceived competence (z) | 0.004 | 0.011 | 0.34 | 0.735 |
| total laughs (count) | 0.000 | 0.001 | 0.47 | 0.639 |
| **colaughter (%)** | **0.136** | **0.047** | **2.90** | **0.004** |

tended to have higher correlation. As we revisit below, this suggests that body synchrony may be partly due to local effects of interaction structure. Laughter can be a shared and pronounced behavioural event, and could drive some of the body correlation signal.

The exploratory model was run without maximizing random effects structure. To confirm that these effects were still present with a more complex triad-level random structure, we ran a follow-up linear mixed effects model with cultural style matching, language matching and laughter. All three variables were significant, again with laughter and language style matching bearing the strongest relationship to observed body synchrony.

In general, we were surprised by the absence of any effects across many of these additional measures. There were neither significant relationships with PD play, nor with perceived warmth or competence. This raises questions about the functional role of overt behavioural synchrony, something we revisit in the General discussion.

## 6.3. Follow-up exploration of body/language interaction

As we discuss below, the relationship between body synchrony and language style matching may reflect a trade-off. That trade-off could be described as 'compensatory,' in the sense that these dynamic processes—moving together or speaking similarly—may operate together to support fluid interaction, but in general when one of them is present, the other need not be. If a triad shares overt body dynamics, it may not need to signal affiliation through language, and so it relaxes that constraint. This predicts a kind of *interaction* between language style matching and body coordination.

To test this idea, we built a simpler model, factoring in only a pair's maximum cross-correlation, and their language style matching. We combined these variables as two fixed factors to predict PD cooperation and perceived warmth, interactive outcome variables, treated as a binomial output variable. It is important to note that this test is based on considerably less data, because the outcome variable is only sampled twice for each pair composing the 35 dyads ($N = 105$ pairs $\times 2$ PD runs $= 210$).

In this follow-up model, we do detect potential relationships between body correlation and language style matching that suggests an interaction, $p < 0.05$. Their main effects are not significant in this model, only their participation in this interaction predicting PD. The effect on cooperating in the game is shown in figure 5. We revisit this in General discussion below, but observe that this may be a so-called 'Goldilocks' pattern. Too much coordination may appear disingenuous, while too little may indicate a failed alliance. Though these are exploratory analyses and we must interpret them with caution, it suggests that measures of pair similarity (such as language, or body movement) cannot be interpreted in simple unidirectional terms—they may be in a trade-off, and may relate more complexity to other interactive variables.

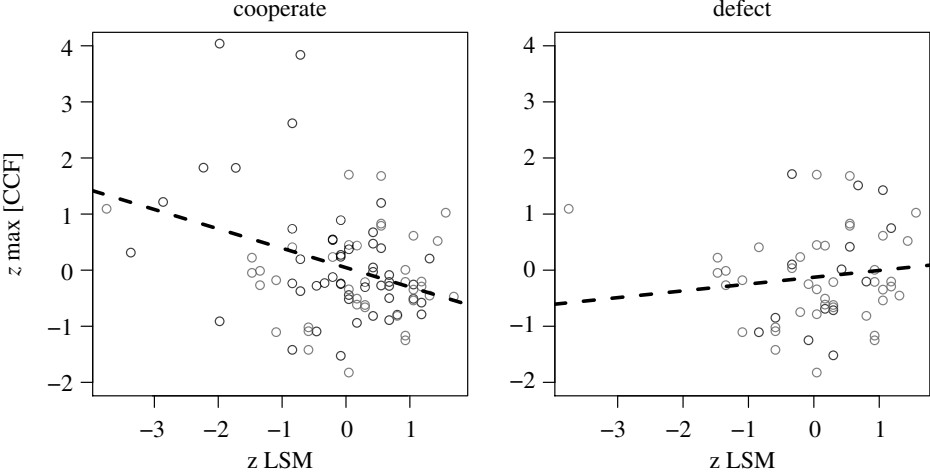

**Figure 5.** To unpack the interaction, we observe in the follow-up model, we plot separately those individuals who cooperated (left) versus defected (right) with their partner. Defections tend to correspond to a line that suggests too little/too much similarity. Cooperation falls on a line that suggests a trade-off, especially for a number of subjects who exhibited strong body synchrony with their partners.

## 6.4. Summary of results

Triads exhibited reliable body correlation, both within their component pairs, but also in the 'ebb and flow' that they exhibited as a group. This correlation is low but significantly greater than a surrogate baseline, built from the correlations of persons across conversations. This small but significant correlation tended to organize itself around a lag of 0. When examining 10 s segments of body correlation, humans sometimes show correlation as high as 0.8–0.9, indicating extremely tightly locked movements; at other times, this correlation can be quite low, even as low as –0.5, indicating that there is quite a range of correlation structure in any 10 s segment.

These data originated in a study examining personality and conversational behaviour that included many additional measures. These measures permitted exploration of the potential functional role for body correlation within the triad. We do not find strong relationships between body correlation and these additional variables. If anything, our exploratory analysis revealed that cultural and language style matching *inversely* correlate with body movement. As we summarize below, this may be due to a more *compensatory* nature of body correlation: it may be serving to smooth out interactions that are experiencing disruption at some other level. Given the exploratory nature of these analyses, and their relatively weak outcomes, we expand cautiously on these theoretical implications below.

## 7. General discussion

Humans exhibit rich correlated structure while they interact. They do so in dance and coordinated group activities [5,52], but also in the structure of our most prosaic joint actions, such as a casual conversation [9,10,15,18]. The current results demonstrate that this is the case in triads. Component dyads show time-locked body correlation patterns. The extent to which these dyads were moving together was also correlated at the triadic levels: the triad was moving together *together*. In both cases, this correlation is quite low, but statistically reliable.

But what is the functional significance of this correlated movement? Our exploratory regression result turned up three potential relationships. Cultural and language style matching related *negatively* with body correlation; colaughter related *positively*. The rest of our variables, including PD outcome, did not relate to how much dyads moved together. Body correlation did not predict cooperation or defection, nor the many other measures. This may be due to sampling resolution. Some of these measures are derived from single-item responses in the post-interaction questions. In prior work, some measures *did* correlate in interesting ways with players' strategies in a PD game. For example, greater speech rate convergence predicted cooperation, as did perceptions and warmth and competence [24,25]. Interestingly though, two variables that are contributing significantly in our model—culture and language style matching—did *not* show a relationship to PD game play in prior

work [43]. In subsequent work, participants watched the videos and generally failed to correctly guess the game play of the conversationalists, strongly suggesting that any reliable cues of cooperation in these conversations are subtle at best [25].

Colaughter was positively associated with triadic synchrony in the current analysis, and was related to PD game play in an earlier analysis with the same conversationalists [24,25]. Specifically, males who engaged in more colaughter, and who produced a greater proportion of colaughs relative to individual laughs, were more likely to cooperate. Moreover, that prior study found a positive association between degree of convergence and colaughter counts: the more people laughed together, the more they converged in speech rate. This dovetails nicely with the current finding that triads who laugh together more appear to move more in synchrony. Laughter is intricately tied to breathing [53], and has been shown to signal various kinds of pragmatic information, including that important for conversational rhythm [54,55]. Listeners from all over the world are quite adept at identifying friends versus strangers from 1 s excerpts of colaughter as well, suggesting that the co-produced signal could potentially signal affiliation information to third parties [56]. The current analysis reveals further the complexity of laughter signals in potentially modulating interactive dynamics at multiple levels of analysis and timescales [18].

## 7.1. Generator versus signature of interaction

The original task on which this corpus is based was a generic conversational prompt, followed by other tasks and surveys. This set-up may provide clues about the relatively weak effects observed. Participants interacted before any other feature of the study was introduced. This means that body correlation could be subject to the *generic* constraints of seeking to interact with other people. As in prior work, this generates a low but reliable level of aggregate synchrony. The present results show that synchrony can be both quite high and low across 10 s segments of the interaction, but when averaged it shows a reliable positive correlation at lag 0 relative to baseline. This could reflect a general structural feature of interactive exchanges. In other words, there was no element to the task instructions other than their interaction that might serve to more clearly relate their dynamics (e.g. as in prior work: argument, puzzle solving, etc.).

Body motion may be participating as a *signature* rather than a *generator* of potential future alliances or affiliation. This predicts that, as a signature, any modification of the task may produce concomitant changes in dynamics (e.g. in the body or language). For example, past work has shown that putting participants in the situation of argument can impact this body correlation signature [10,11]. The variance produced in a context such as client–therapist interactions may elicit a stronger signature of the alliance produced between them [57]. The same may be said for potential romantic partners—the increased variance associated with outcomes in a potential romantic encounter may render the body signature stronger in its functional correlations (cf. [15]). Thus, with a more structured task goal, we may have obtained different results. In casual conversation, the signature from body movement may simply be too weak to discern potential alliance formation.

An important limitation of the present work is that the triads are captured in the same shot. This means that the conversant in the middle position does not reveal forward/back body movement as much as would those on the left and right. While it is likely that most body movement will displace pixels in the video, even for the middle conversant, it may nevertheless mean a lower measurement sensitivity as well. Expanding measures and improving sensitivity in future analyses may come from other multi-person tasks that involve verbal and non-verbal designs (e.g. [40,58,59]). The aim of the present research was to integrate motion signals from admittedly coarse video processing with a rich set of interactive covariates. Higher-quality motion-capture data and other multimodal methods in the future may yield data sources that overcome the need for aggressive filtering and lend more confidence to segment-wise motion, such as facial expression, hand movement or nodding (e.g. [60,61]).

There is also promise in improving sensitivity by adapting time-series analysis methods that focus on subtler features and relationships among body motion signals. For example, Dean & Dunsmuir [47] suggest linear transfer function models, and work by Irvin and colleagues [62] suggests that nonlinear causal analyses, like convergent cross mapping (CCM), offers unique insights. It may be valuable to decompose frequency components of these interactions such as through wavelet methods [63], or even to find explicit multi-person measures, some of them deriving from nonlinear analyses of coupled systems like recurrence quantification [37,38] or cluster-phase methods [31]. Here we aimed to test perhaps the most generic cross-correlation method, but it is clear that future work should seek to systematize and codify where and when these more advanced methods may be relevant (cf. [46]).

These analysis methods may also enhance exploration of the role of physical configuration relevant here. In particular, causal analyses may determine if an interlocutor occupying a centre position drives more of the conversational dynamic, or if the more aligned face-to-face configuration of the left and right participants dominates these dynamics. The important structuring influence of physical configuration has long been remarked in research on face-to-face human communication [64,65].

It is important to note that the number of possible analyses that can be conducted on multiple time series is rather large, and this poses a general challenge for the field of human communication. This challenge is isolating the most fruitful measurements that contain signal regarding the progress or outcome of an interaction. Here we started with the simplest thesis: signal may derive from time-aligned covariation of gross body movement between three interacting humans. Our results offer some suggestive patterns, but do not yield clear answers. The number of potential follow-up analyses is quite large: causal and leader/follower dynamics, temporal changes across the course of interaction, the presence or absence of particular modes or events in interaction, and so on. It is possible that the subtle uniqueness of any one interaction cannot support what would be regarded as stable principles because interactive behaviours specifically, and cognition in general, are massively contextually labile (cf. [66,67]).

It is also possible that synchrony is not serving a communicative signal, but indexing something else. For example, the 'weirdness' of the interaction: in a tight, well-lit space, face-to-face with strangers, with the camera rolling, perhaps few participants want to draw attention to themselves by moving. Pronounced movements might be suppressed by the awkwardness of the exchange, but people still need to shift around. When one person moves, others can take the opportunity to move, similar to the dynamics of other disruptive phenomena such as coughing in the classroom [68]. Such shared states may 'license' more relaxed body movements, and the correlation structure in the body may index their satisfaction with the *task* overall, because participants who license each other to move more (more correlation) may have found the task less stressful. The positive relationship with laughter supports this more 'local' explanation, because laughter could also facilitate defusing the situation. Careful task analysis of this sort would certainly help unpack the function of shared bodily dynamics, but they would require additional outcome measures in future work.

## 7.2. 'Goldilocks' phenomenon?

But what of the relatively stronger negative relationship with language style matching? Past work may help to explain this too. Some have characterized a *compensatory* role for the correlation of overt signals in human interaction. For example, when participants perceive a breakdown in communication, they may enhance the correlation of their speech and eye movements [16,69]. When participants perceive a task partner as a member of an opposing group, bodily synchrony may be increased [30]. Body motion may thus be *participating* with language and perceived cultural fit to help smooth out the general structure of the interaction. If there is a breakdown in one end of the interaction, such as lower language matching, then body motion correlation may compensate.

Though a promising theoretical avenue, this admittedly does not explain the lack of relationship of body movement with perceived interruptions and warmth and competence. The sort of data we analyse here is unusual—an array of rich signals coupled to a variety of functionally relevant additional measures. If this dataset is reflective of general patterns of casual interaction, then there is indeed much left to explain, and perhaps some concern. The signature from the body may be present, but too weak as a signature of affiliative or alliance-building outcome in these more casual, prosaic tasks (especially, perhaps, in triads and larger groups).

The interaction that is present between body movement and language style matching does predict PD cooperation, albeit weakly. The pattern of data bears resemblance to a 'Goldilocks' phenomenon. Too much shared behaviour (high $z$-scores in figure 5) could reflect dishonesty or disingenuousness. Too little shared behaviour (low $z$-scores in figure 5) could reflect a failed alliance. A more careful balance between these behaviours (line in figure 5, left), may reflect a 'just right' combination of behaviours to support cooperation. Again, we need to interpret the results cautiously here given the smaller amount of data in the PD decisions, but the pattern of results is sufficiently interesting to motivate future work.

## 7.3. Synergies and interaction

We tentatively note that this potential 'trade-off' between cultural, language and body mutuality may be reflective of interacting systems seeking efficient communication. This general theoretical observation should be made with caution, but we offer it here: body correlation and at least two other

behavioural signals are not merely mutually reinforcing, but rather show an inverse relationship that suggests one or the other may fill a particular interactive function. This relates to the important observations of Fusaroli and colleagues [34,36]. They recommend the concept of 'synergy' to explain the dynamics of human interaction (see also [37,38,70,71]). Interacting persons do not merely amplify their component behaviours. Instead, they strike a functional balance that might reflect a kind of stabilization of behaviour across a range of potential solutions to interaction (cf. [14,72]). There are many ways that we can successfully structure our interactions. Humans must find a combination of behaviours that succeeds in doing so without losing flexibility and the potential for rapid continuation of an exchange. The result is a kind of *balance* among components. This predicts that interactive behaviours ought to exhibit occasional trade-offs. The role played by language style matching may obviate any contributions by the body and vice versa. Though the negative correlations we observe suggest this, the weak outcome of our exploratory models recommend caution but, to us, excitement in seeking follow-up analyses in other datasets and experimentation.

# 8. Conclusion

Triadic conversants engage in synchrony. The functional reason for it is not entirely clear. It could be an epiphenomenon of general human interaction. Perhaps it reflects a generic balancing among mutual behaviours in interaction. The functional role may be found in other analysis strategies not considered here. The role of body correlation could emerge from the more local aspects of an exchange. Perhaps a joke leads to a sharp 10 s window of mutual body entrainment, with laughter and other accompaniments. At other times, an intense recounting of stress in our triads may elicit the opposite: gesticulation in the storyteller, and staid listening in their partner. Future work should unpack these more local relations, as they might explain the body correlation observed here and elsewhere.

Ethics. All procedures used for original data collection were approved by the UCLA Institutional Review Board (#G07-10-097-01).

Data accessibility. All analysis scripts, along with raw data, can be found at https://github.com/racdale/triadic-bodily-synchrony.

Authors' contributions. M.M.G. and J.H.M. collected the corpus; M.M.G., J.H.M. and G.A.B. conducted prior analysis and data assistance; R.D. and G.A.B. planned the present re-analysis; R.D. developed the scripts and conducted analysis; R.D., G.A.B., J.H.M. and M.M.G. wrote the paper.

Competing interests. The authors declare no competing interests.

Funding. Support for original data collection and analysis provided by an International Society for Human Ethology grant to M.M.G., and a UCLA Faculty Research grant to J.H.M.

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
