## [Reviewer comments · Royal Society Open Science]

Review History

RSOS-200095.R0 (Original submission)

Review form: Reviewer 1

Is the manuscript scientifically sound in its present form?

Yes

Are the interpretations and conclusions justified by the results?

Yes

Is the language acceptable?

Yes

Do you have any ethical concerns with this paper?

No

Have you any concerns about statistical analyses in this paper?

Yes

Recommendation?

Accept with minor revision (please list in comments)

Comments to the Author(s)

The work is clearly presented and rigorous, and builds upon a solid body of work in the field, including by the authors themselves. The main contribution of the work concerns triadic interaction. The work builds up to this by first studying the constituent dyads, and then combining these.

Highlighting some limitations on the analysis might be helpful. For example, what is the impact of the fact that the movement data is obtained from a 2D representation of a 3D situation? Data on left and right participants reveals data on backwards and forwards movements, while the data on the centre participant would seem to reveal more about their sideways movements. I'm also curious about the location of the participants, and their relation to the camera. Does this have any impact on their roles within interactions – does the centre participant tend to be more dominant, or more involved in the conversations, than either of the other two, for example?

There are places in the manuscript where information is a little sparse and would benefit from expansion. Because the dataset is taken from a prior study (Gervais et al., 2013), much of the description of the procedure is left to citation. However, it would be helpful towards the reader to include a brief summary of that prior work, how it differs from the current paper, and some further detail on the procedures used. The Prisoner's Dilemma (PD) result, for example, is referred to, yet the specifics of how this was carried out are a little vague (e.g. I assume this was a 3-person PD?) – this is easily solved by a single-sentence description of the task.

Technically, the manuscript would benefit from some additional details. The 2nd-order cross-correlation function, for example, is not clearly described. The term is one used in physics, I believe, but seems differently employed here. On first reading I assumed a cross-correlation applied to the output correlations of the pairwise, dyadic cross-correlations. It is not immediately clear if this applied to all 3 possible combinations of dyads, or just 2 dyads. Looking at the provided code in R, this seems to be an average of all 3 pairwise correlations – is this effectively the same thing? How valid is this as an approach? It would be good to include some further explanation and justification of the approach used.

The exploratory nature of the work allows the authors a lot of freedom with the analysis, and they clearly acknowledge this throughout. However, I would have liked the exploration to go a bit deeper on both the dynamics of the interactions. One analysis that I feel is missing would be on the leader-follower roles within conversations – what does the cross-correlation approach reveal about who is talking, and who is listening, and what kind of synchrony patterns are observed during turn-taking.

A further analysis concerns the changes in synchrony over the 10 minutes of conversation – how does the conversation duration affect the way people interact?

How such dynamic considerations might scale from dyads to triads is an open question, and one that would have been good to see discussed here – even if only as a pointer to future work.

The paper is well-written and follows a logical structure. There are, however, a handful of places where the text could be a bit clearer – or include additional information.

The prior work is referred to later in the manuscript, but in a slightly ambiguous manner – on p23, for example, it is unclear whether the authors are referring to (Gervais et al., 2013) or (Manson et al., 2013). Any references comparing findings from prior work should ideally be more explicit.

Additionally, the first sentence of “Generator vs. signature of interaction” (p22), refers to sparse results – but it is unclear on first reading which result in particular is ‘sparse’. Also, the structure of the task might not be entirely clear in the reader's mind, so it could be worth re-stating directly. Also, the next sentence then begins, ‘First...’ but there is not ‘Second...’ to match up with it later on.

On p7, an 8th-order Butterworth filter is described with a low-frequency cut-off of 0.05 Nyquist. Please explicitly state this frequency and how it is arrived at (e.g., I'm guessing something like $30 \text{ fps} * 0.5 * 0.05 = 0.75 \text{ Hz}$, or a period of 1.33s?)

On p9, the phrase 'move in the same way' is used. However, from my understanding, the way in which people move (suggesting some spatial coordination) is irrelevant – the measure is primarily on the synchrony (and amount) of movement?

Table 1 is missing information on sex – why?

Figure 5 might be better explained - re-state what the axes show in the caption. It could be me, but I need some more information on the jittering process within the main body of the text. Also green and red are not ideal differentiators (a common confusion for colour blind people).

Review form: Reviewer 2

Is the manuscript scientifically sound in its present form?

Yes

Are the interpretations and conclusions justified by the results?

Yes

Is the language acceptable?

Yes

Do you have any ethical concerns with this paper?

No

Have you any concerns about statistical analyses in this paper?

Yes

Recommendation?

Major revision is needed (please make suggestions in comments)

Comments to the Author(s)

The authors aim at testing for the role of behavioral interaction in triadic tasks. The paper is generally well written, and investigates two important topics of interest, namely how joint action coordination behaves differently in larger groups compared to dyads, and what role behavioral synchrony actually serves. The paper is interesting, but I have a couple of methodological issues listed below that the authors need to address.

p.6 It remained unclear which parts of the body were tracked, whether these were different for different groups (or within a group), and which effects this might have on the synchronisation measures.

p.7 Please provide the absolute frequency cut-off used, not relative to the Nyquist frequency. Otherwise, it is difficult to judge the reasonability of the cut-off criterion chosen.

p.9 Using cross-correlation because it is „its computation is easy to understand and output relatively transparent“ is probably not the best reason to use this method. The question is, whether it adequately captures the properties of the data, and whether its assumptions are met.

Hence, the authors should definitely check for violations of assumptions - linearity, distribution of residuals, presence of outliers - and provide the outcome of the assumption tests in the paper.

p.12 There is something odd about table 1 for the row „sex“ ...

p.12 I am sorry if I missed this, but how was „triadic correlation“ calculated?

p.13 Please describe in more detail, perhaps using an example, how exactly second-order cross-correlation (i.e., the triadic correlation?) was computed. I am not sure whether I correctly understand it, and whether this measure seems to make sense as the label suggests.

p. 14 Please specify the equation for your MLM. It is hard to tell from your description what was included there and how. For example, if you treat your participant's data as time series, you might want to include an AR1 residual or similar predictor, but it is not clear whether you did so or not.

p. 14 I think it is totally fine to do an exploratory analysis and not correct for multiple testing, but you should still state the level of significance you are setting.

In the discussion, I am missing how methodological shortcomings might have resulted in unexpected patterns of effects - null effects and negative effects. It might be, that the time series are not adequately modelled in terms of linear functions (see my point about assumption tests above), or that triadic interactions are not adequately captured by component dyadic interactions (see Wallot, S., Roepstorff, A., & Mønster, D. (2016). Multidimensional Recurrence Quantification Analysis (MdrQA) for the analysis of multidimensional time-series: A software implementation in MATLAB and its application to group-level data in joint action. *Frontiers in psychology*, 7, 1835.), or that the way motion capturing of certain body regions at the expense of others did simply not measure the relevant dimensions of movement... Given the unexpected results and the exploratory nature of the study, these might be other potential sources of what could and could not be observed here.

Decision letter (RSOS-200095.R0)

10-Mar-2020

Dear Dr Dale,

The editors assigned to your paper ("Body synchrony in triadic interaction") have now received comments from reviewers. We would like you to revise your paper in accordance with the referee and Associate Editor suggestions which can be found below (not including confidential reports to the Editor). Please note this decision does not guarantee eventual acceptance.

Please submit a copy of your revised paper before 02-Apr-2020. Please note that the revision deadline will expire at 00.00am on this date. If we do not hear from you within this time then it will be assumed that the paper has been withdrawn. In exceptional circumstances, extensions may be possible if agreed with the Editorial Office in advance. We do not allow multiple rounds of revision so we urge you to make every effort to fully address all of the comments at this stage. If deemed necessary by the Editors, your manuscript will be sent back to one or more of the original reviewers for assessment. If the original reviewers are not available, we may invite new reviewers.

To revise your manuscript, log into <http://mc.manuscriptcentral.com/rsos> and enter your Author Centre, where you will find your manuscript title listed under "Manuscripts with

Decisions." Under "Actions," click on "Create a Revision." Your manuscript number has been appended to denote a revision. Revise your manuscript and upload a new version through your Author Centre.

- Data accessibility

If you wish to submit your supporting data or code to Dryad (<http://datadryad.org/>), or modify your current submission to dryad, please use the following link:
<http://datadryad.org/submit?journalID=RSOS&manu=RSOS-200095>

- Competing interests

- Authors' contributions

- Acknowledgements

- Funding statement

Kind regards,

Andrew Dunn

on behalf of Dr César Lima (Associate Editor) and Essi Viding (Subject Editor)

Comments to Author:

Reviewers' Comments to Author:

Reviewer: 1

Comments to the Author(s)

The work is clearly presented and rigorous, and builds upon a solid body of work in the field, including by the authors themselves. The main contribution of the work concerns triadic interaction. The work builds up to this by first studying the constituent dyads, and then combining these.

Highlighting some limitations on the analysis might be helpful. For example, what is the impact of the fact that the movement data is obtained from a 2D representation of a 3D situation? Data on left and right participants reveals data on backwards and forwards movements, while the data on the centre participant would seem to reveal more about their sideways movements. I'm also curious about the location of the participants, and their relation to the camera. Does this have any impact on their roles within interactions – does the centre participant tend to be more dominant, or more involved in the conversations, than either of the other two, for example?

There are places in the manuscript where information is a little sparse and would benefit from expansion. Because the dataset is taken from a prior study (Gervais et al., 2013), much of the description of the procedure is left to citation. However, it would be helpful towards the reader to include a brief summary of that prior work, how it differs from the current paper, and some further detail on the procedures used. The Prisoner's Dilemma (PD) result, for example, is referred to, yet the specifics of how this was carried out are a little vague (e.g. I assume this was a 3-person PD?) – this is easily solved by a single-sentence description of the task.

Technically, the manuscript would benefit from some additional details. The 2nd-order cross-correlation function, for example, is not clearly described. The term is one used in physics, I believe, but seems differently employed here. On first reading I assumed a cross-correlation applied to the output correlations of the pairwise, dyadic cross-correlations. It is not immediately clear if this applied to all 3 possible combinations of dyads, or just 2 dyads. Looking at the provided code in R, this seems to be an average of all 3 pairwise correlations – is this effectively the same thing? How valid is this as an approach? It would be good to include some further explanation and justification of the approach used.

The exploratory nature of the work allows the authors a lot of freedom with the analysis, and they clearly acknowledge this throughout. However, I would have liked the exploration to go a bit deeper on both the dynamics of the interactions. One analysis that I feel is missing would be on the leader-follower roles within conversations – what does the cross-correlation approach reveal about who is talking, and who is listening, and what kind of synchrony patterns are observed during turn-taking.

A further analysis concerns the changes in synchrony over the 10 minutes of conversation – how does the conversation duration affect the way people interact?

How such dynamic considerations might scale from dyads to triads is an open question, and one that would have been good to see discussed here – even if only as a pointer to future work.

The paper is well-written and follows a logical structure. There are, however, a handful of places where the text could be a bit clearer – or include additional information.

The prior work is referred to later in the manuscript, but in a slightly ambiguous manner – on p23, for example, it is unclear whether the authors are referring to (Gervais et al., 2013) or (Manson et al., 2013). Any references comparing findings from prior work should ideally be more explicit.

Additionally, the first sentence of “Generator vs. signature of interaction” (p22), refers to sparse results – but it is unclear on first reading which result in particular is ‘sparse’. Also, the structure of the task might not be entirely clear in the reader’s mind, so it could be worth re-stating directly. Also, the next sentence then begins, ‘First...’ but there is not ‘Second...’ to match up with it later on.

On p7, an 8th-order Butterworth filter is described with a low-frequency cut-off of 0.05 Nyquist. Please explicitly state this frequency and how it is arrived at (e.g., I’m guessing something like $30 \text{ fps} * 0.5 * 0.05 = 0.75 \text{ Hz}$, or a period of 1.33s?)

On p9, the phrase ‘move in the same way’ is used. However, from my understanding, the way in which people move (suggesting some spatial coordination) is irrelevant – the measure is primarily on the synchrony (and amount) of movement?

Table 1 is missing information on sex – why?

Figure 5 might be better explained - re-state what the axes show in the caption. It could be me, but I need some more information on the jittering process within the main body of the text. Also green and red are not ideal differentiators (a common confusion for colour blind people).

Reviewer: 2

Comments to the Author(s)

The authors aim at testing for the role of behavioral interaction in triadic tasks. The paper is generally well written, and investigates two important topics of interest, namely how joint action coordination behaves differently in larger groups compared to dyads, and what role behavioral synchrony actually serves. The paper is interesting, but I have a couple of methodological issues listed below that the authors need to address.

p.6 It remained unclear which parts of the body were tracked, whether these were different for different groups (or within a group), and which effects this might have on the synchronisation measures.

p.7 Please provide the absolute frequency cut-off used, not relative to the Nyquist frequency. Otherwise, it is difficult to judge the reasonability of the cut-off criterion chosen.

p.9 Using cross-correlation because it is „its computation is easy to understand and output relatively transparent“ is probably not the best reason to use this method. The question is, whether it adequately captures the properties of the data, and whether its assumptions are met. Hence, the authors should definitely check for violations of assumptions - linearity, distribution of residuals, presence of outliers - and provide the outcome of the assumption tests in the paper.

p.12 There is something odd about table 1 for the row „sex“ ...

p.12 I am sorry if I missed this, but how was „triadic correlation“ calculated?

p.13 Please describe in more detail, perhaps using an example, how exactly second-order cross-correlation (i.e., the triadic correlation?) was computed. I am not sure whether I correctly understand it, and whether this measure seems to make sense as the label suggests.

p. 14 Please specify the equation for your MLM. It is hard to tell from your description what was included there and how. For example, if you treat your participant's data as time series, you might want to include an AR1 residual or similar predictor, but it is not clear whether you did so or not.

p. 14 I think it is totally fine to do an exploratory analysis and not correct for multiple testing, but you should still state the level of significance you are setting.

In the discussion, I am missing how methodological shortcomings might have resulted in unexpected patterns of effects - null effects and negative effects. It might be, that the time series are not adequately modelled in terms of linear functions (see my point about assumption tests above), or that triadic interactions are not adequately captured by component dyadic interactions (see Wallot, S., Roepstorff, A., & Mønster, D. (2016). Multidimensional Recurrence Quantification Analysis (MdrQA) for the analysis of multidimensional time-series: A software implementation in MATLAB and its application to group-level data in joint action. *Frontiers in psychology*, 7, 1835.), or that the way motion capturing of certain body regions at the expense of others did simply not measure the relevant dimensions of movement... Given the unexpected results and the exploratory nature of the study, these might be other potential sources of what could and could not be observed here.

Author's Response to Decision Letter for (RSOS-200095.R0)

See Appendix A.

RSOS-200095.R1 (Revision)

Review form: Reviewer 1

Is the manuscript scientifically sound in its present form?

Yes

Are the interpretations and conclusions justified by the results?

Yes

Is the language acceptable?

Yes

Do you have any ethical concerns with this paper?

No

Have you any concerns about statistical analyses in this paper?

Yes

Recommendation?

Accept with minor revision (please list in comments)

Comments to the Author(s)

This paper is a lovely read, and although the work is largely exploratory, it includes some interesting findings worthy of publication. I thank the authors for their detailed responses to my earlier comments. The issues raised in my initial review were mostly met. However, on closer inspection of the revised draft, I would like to request some further clarifications.

The description of the 2nd-order cross correlation is somewhat clearer now, however I fear it does not go far enough. The new text states that you take the windowed cross-correlations from each of the 3 constituent dyads, you then "calculate a cross correlation" from these. How exactly is this 3-way cross correlation calculated, specifically, how are the 3 cross-correlation graphs combined? (i.e. is there another round of pairwise cross-correlations that then get averaged? Or are the 3 cross-correlations averaged directly?) A diagram would help enormously. It might help to add a specific worked example, perhaps using a fuller expansion of Figure 3 to include the 2nd-order stage?

When presenting results using triads, each composed of 3 dyads, there is inevitably room for confusion. Consequently there are several places in the paper where it can be hard to work out exactly what is being compared, particularly when trying to interpret the statistics.

The value of N, for example, should be made available for every evaluation. A little bit more information might help clear up potential confusion. Specifically, please specify how many dyads are used when calculating the dyadic statistics (e.g. for Tables 1 and 2). If the number of triads is 35, then I assume the number of dyads considered is 3×35 ? However, the text seems to indicate $N=35$. In the PD exploratory study (p21), the number of pairs is given as $N=105$. How is this number arrived at, particularly when there are only 105 participants in total?

The repeated measures approach, too, raises some questions. How many 10s windows are used in analysing each dyad/triad? Do these windows overlap (i.e. are they mutually exclusive)? It would also be good to clarify exactly how the windows move. Do they roll forward one raw data sample at a time, or do they jump by a full window-length? Finally, please give some reasoning on why you chose a duration of 10s.

The remainder of my comments concern minor corrections, typos and clarifications.

p6.18 - 'we predict a variety of individual differences and outcome measures...' - it would be cleaner if this was more specific, i.e. state how many predictions, followed by a concise list.

p6.29 - 'This includes _a_ detailed summary...'

p8.46 - '...motion signal_s_ from similar...'

p8.46 - '...0.2_Hz_ cutoff...' (missing units)

p8.55 - '...before analyzing_the_ main results...'

p10.36 - Briefly justify why 10s. Also, is this a rolling or jumping (non-overlapping) window?

p14.Table 1 - N value(s)?

p16.27-39 - '... a few predicted outcomes...'. This relates to my first comment on the introduction: please enumerate and be more specific about these outcomes.

p17.18-24 - 'Dyads showed sig... $p < .0001$ ' this sentence is unclear. It opens talking about dyads, but then introduces a one-sampled t-test on the triads. Please unpick and clarify.

p17.Table 2 - N value(s)?

p18.Figure 4 - middle figure is wrong: the y-axis states 'number of dyads', yet the distribution sums up to 100 (i'm assuming $N=35$, and the distribution is percentages? Please clarify.) A legend indicating observed/surrogate would be helpful on these plots.

p19.3 - 'r=.15'... then later 'r=.16'... a little inconsistent

p20.Table 3 - it might be helpful to specify 'unstandardized beta (B)', 'standard error (SE)', etc. either in the table or in the text.

p21.47 - $N=105$ pairs... yet earlier $N=105$ participants. Clarify.

Review form: Reviewer 2

Is the manuscript scientifically sound in its present form?

Yes

Are the interpretations and conclusions justified by the results?

Yes

Is the language acceptable?

Yes

Do you have any ethical concerns with this paper?

No

Have you any concerns about statistical analyses in this paper?

No

Recommendation?

Accept as is

Comments to the Author(s)

Dear authors, thank you for the thorough revision - I recommend publication of the manuscript.

Decision letter (RSOS-200095.R1)

Dear Dr Dale:

On behalf of the Editors, I am pleased to inform you that your Manuscript RSOS-200095.R1 entitled "Body synchrony in triadic interaction" has been accepted for publication in Royal Society Open Science subject to minor revision in accordance with the referee suggestions. Please find the referees' comments at the end of this email.

The reviewers and Subject Editor have recommended publication, but also suggest some minor revisions to your manuscript. Therefore, I invite you to respond to the comments and revise your manuscript.

- Ethics statement

- Data accessibility

If you wish to submit your supporting data or code to Dryad (<http://datadryad.org/>), or modify your current submission to dryad, please use the following link:
<http://datadryad.org/submit?journalID=RSOS&manu=RSOS-200095.R1>

- Competing interests

- Authors' contributions

AB carried out the molecular lab work, participated in data analysis, carried out sequence alignments, participated in the design of the study and drafted the manuscript; CD carried out

the statistical analyses; EF collected field data; GH conceived of the study, designed the study, coordinated the study and helped draft the manuscript. All authors gave final approval for publication.

- Acknowledgements

- Funding statement

Because the schedule for publication is very tight, it is a condition of publication that you submit the revised version of your manuscript before 29-Jul-2020. Please note that the revision deadline will expire at 00.00am on this date. If you do not think you will be able to meet this date please let me know immediately.

Kind regards,

Anita Kristiansen
Editorial Coordinator

on behalf of Dr César Lima (Associate Editor) and Essi Viding (Subject Editor)
openscience@royalsociety.org

Reviewer comments to Author:
Reviewer: 2

Comments to the Author(s)
Dear authors, thank you for the thorough revision - I recommend publication of the manuscript.

Reviewer: 1

Comments to the Author(s)
This paper is a lovely read, and although the work is largely exploratory, it includes some interesting findings worthy of publication. I thank the authors for their detailed responses to my earlier comments. The issues raised in my initial review were mostly met. However, on closer inspection of the revised draft, I would like to request some further clarifications.

The description of the 2nd-order cross correlation is somewhat clearer now, however I fear it does not go far enough. The new text states that you take the windowed cross-correlations from each of the 3 constituent dyads, you then "calculate a cross correlation" from these. How exactly is this 3-way cross correlation calculated, specifically, how are the 3 cross-correlation graphs combined? (i.e. is there another round of pairwise cross-correlations that then get averaged? Or are the 3 cross-correlations averaged directly?) A diagram would help enormously. It might help to add a specific worked example, perhaps using a fuller expansion of Figure 3 to include the 2nd-order stage?

When presenting results using triads, each composed of 3 dyads, there is inevitably room for confusion. Consequently there are several places in the paper where it can be hard to work out exactly what is being compared, particularly when trying to interpret the statistics.

The value of N, for example, should be made available for every evaluation. A little bit more information might help clear up potential confusion. Specifically, please specify how many dyads are used when calculating the dyadic statistics (e.g. for Tables 1 and 2). If the number of triads is 35, then I assume the number of dyads considered is 3×35 ? However, the text seems to indicate $N=35$. In the PD exploratory study (p21), the number of pairs is given as $N=105$. How is this number arrived at, particularly when there are only 105 participants in total?

The repeated measures approach, too, raises some questions. How many 10s windows are used in analysing each dyad/triad? Do these windows overlap (i.e. are they mutually exclusive)?

It would also be good to clarify exactly how the windows move. Do they roll forward one raw data sample at a time, or do they jump by a full window-length? Finally, please give some reasoning on why you chose a duration of 10s.

The remainder of my comments concern minor corrections, typos and clarifications.

p6.18 - 'we predict a variety of individual differences and outcome measures...' - it would be cleaner if this was more specific, i.e. state how many predictions, followed by a concise list.

p6.29 - 'This includes _a_ detailed summary...'

p8.46 - '...motion signal_s_ from similar...'

p8.46 - '...0.2_Hz_ cutoff...' (missing units)

p8.55 - '...before analyzing _the_ main results...'

p10.36 - Briefly justify why 10s. Also, is this a rolling or jumping (non-overlapping) window?

p14.Table 1 - N value(s)?

p16.27-39 - '... a few predicted outcomes...'. This relates to my first comment on the introduction: please enumerate and be more specific about these outcomes.

p17.18-24 - 'Dyads showed sig... $p < .0001$ ' this sentence is unclear. It opens talking about dyads, but then introduces a one-sampled t-test on the triads. Please unpick and clarify.

p17.Table 2 - N value(s)?

p18.Figure 4 - middle figure is wrong: the y-axis states 'number of dyads', yet the distribution sums up to 100 (i'm assuming $N=35$, and the distribution is percentages? Please clarify.) A legend indicating observed/surrogate would be helpful on these plots.

p19.3 - 'r=.15'... then later 'r=.16'... a little inconsistent

p20.Table 3 - it might be helpful to specify 'unstandardized beta (B)', 'standard error (SE)', etc. either in the table or in the text.

p21.47 - $N=105$ pairs... yet earlier $N=105$ participants. Clarify.

Author's Response to Decision Letter for (RSOS-200095.R1)

See Appendix B.

Decision letter (RSOS-200095.R2)

Dear Dr Dale,

It is a pleasure to accept your manuscript entitled "Body synchrony in triadic interaction" in its current form for publication in Royal Society Open Science.

on behalf of Dr César Lima (Associate Editor) and Essi Viding (Subject Editor)
openscience@royalsociety.org

Appendix A

Dear Drs. Lima and Viding,

We have now extensively revised our manuscript thanks to the very helpful suggestions from reviewers. Below we detail how we accommodated every point for each reviewer, and include pointers to those revised sections of our manuscript. We thank you for this opportunity to be considered for publication in *Royal Society Open Science*, and look forward to your response.

Best wishes,

Rick and co-authors

Reviewer 1

The work is clearly presented and rigorous, and builds upon a solid body of work in the field, including by the authors themselves. The main contribution of the work concerns triadic interaction. The work builds up to this by first studying the constituent dyads, and then combining these.

We thank the reviewer for their positive remarks, and constructive feedback. We've accommodated all suggestions, all with direct changes to our manuscript.

Highlighting some limitations on the analysis might be helpful. For example, what is the impact of the fact that the movement data is obtained from a 2D representation of a 3D situation?

This is an important point and we now acknowledge it in the General Discussion, in the section added that begins, "Indeed, an important limitation of the present work is that the triads are captured in the same shot..." (near p. 26) We note that while the middle conversant likely leads to lower measurement sensitivity, pixels are likely still displaced with body movement, and our correlation measures are derived from gross body movement (rather than along any particular axis of the 2D video). Still, it's an important limitation and we now acknowledge this in that section. Relatedly:

Data on left and right participants reveals data on backwards and forwards movements, while the data on the centre participant would seem to reveal more about their sideways movements. I'm also curious about the location of the participants, and their relation to the camera. Does this have any impact on their roles within interactions — does the centre participant tend to be more dominant, or more involved in the conversations, than either of the other two, for example?

Just after that same spot referenced above (near p. 26, beginning "Expanding measures and improving sensitivity in future analyses may..."), we add a note about future directions that this may be a valuable contribution to interaction configuration in the future. In particular, it may be possible to use causal analyses to determine if (say) a middle interlocutor drives more of the conversational dynamic, versus (quite possibly) the more aligned face-to-face configuration of the left and right participants instead. This is also described in conjunction with the point about future directions in correlation-based approaches as well (see below).

There are places in the manuscript where information is a little sparse and would benefit from expansion. Because the dataset is taken from a prior study (Gervais et al., 2013), much of the description of the procedure is left to citation. However, it would be helpful towards the reader to include a brief summary of that prior work, how it differs from the current paper, and some further detail on the procedures used.

The Prisoner's Dilemma (PD) result, for example, is referred to, yet the specifics of how this was carried out are a little vague (e.g. I assume this was a 3-person PD?) — this is easily solved by a single-sentence description of the task.

The source material is taken from the original Gervais et al. (2013) paper, and several datasets and subsequent papers have resulted. To convey the study in more detail, we have now added more details regarding the original research, and elaboration of the PD procedure in particular (near p. 6, first paragraph under “Methods”).

Technically, the manuscript would benefit from some additional details. The 2nd-order cross-correlation function, for example, is not clearly described. The term is one used in physics, I believe, but seems differently employed here. On first reading I assumed a cross-correlation applied to the output correlations of the pairwise, dyadic cross-correlations. It is not immediately clear if this applied to all 3 possible combinations of dyads, or just 2 dyads. Looking at the provided code in R, this seems to be an average of all 3 pairwise correlations — is this effectively the same thing? How valid is this as an approach? It would be good to include some further explanation and justification of the approach used.

We now offer a detailed definition of this second-order cross-correlation near p. 11, in the section beginning “To assess whether there is *triadic* synchrony...” The analysis is intended to be a simple, linear extension of first-order correlation. We define “second-order” correlation as a correlation between two window-wise correlation series. In other words, the windowed cross-correlation shown in Fig. 3 (left) is then correlated. We conduct cross-correlation of *all* pairwise windowed measures of synchrony, so the second-order cross-correlation is a sort of average triadic synchrony. If the component dyads are covarying similarly, then this second-order correlation should also show a peak at lag 0 higher than the surrogate baseline. We also added a note in Fig. 3 caption.

The exploratory nature of the work allows the authors a lot of freedom with the analysis, and they clearly acknowledge this throughout. However, I would have liked the exploration to go a bit deeper on both the dynamics of the interactions. One analysis that I feel is missing would be on the leader-follower roles within conversations — what does the cross-correlation approach reveal about who is talking, and who is listening, and what kind of synchrony patterns are observed during turn-taking.

This is a great point and we acknowledge that more detail about these patterns is important. In general, the main dynamics available to us via raw data is simply the optic flow of their interactions. We have opted in the paper to focus our analysis on time-aligned covariation as a common signal of synchrony, and are concerned that seeking additional analyses, especially across so many dyads and across three chair positions, may complicate the overall structure of our study. In lieu of expanding this beyond the present scope, we now acknowledge this important point and elaborate on potential future analyses to make this possible. In addition, the section noted above (p. 26) makes reference to the theoretical relevance as well (e.g., by citing Kendon and the concept of the F-formation). We hope this serves to address this important suggestion. We also note that more sophisticated multimodal datasets may be capable of assisting with this kind of follow-up, and cite several promising directions (p. 26).

A further analysis concerns the changes in synchrony over the 10 minutes of conversation — how does the conversation duration affect the way people interact? How such dynamic considerations might scale from dyads to triads is an open question, and one that would have been good to see discussed here — even if only as a pointer to future work.

We also add a point about temporal progression of correlation on pp. 26-27 (paragraph beginning “Indeed, it is important note that the number of possible analyses that can be conducted on multiple time series is rather large”). We appreciate these points, and acknowledge that there are certain limitations to the present study that need to be reconciled in future analysis. It is true, of course, that the number of possible analyses that can be conducted (temporal change, leader/follower, causal analysis, multimodal, etc.) are so large that we must acknowledge the importance of isolating critical loci of signal in future work. This is now acknowledged in detail on p. 27, and we take our study as a point that it is critical to expand beyond the most common, simplest measures.

The prior work is referred to later in the manuscript, but in a slightly ambiguous manner — on p23, for example, it is unclear whether the authors are referring to (Gervais et al., 2013) or (Manson et al., 2013). Any references comparing findings from prior work should ideally be more explicit.

We have clarified which studies are being referred to in the discussion by expanding our summary (near p. 23, near section starting “In prior work, some measures *did* correlate in interesting...”).

Additionally, the first sentence of “Generator vs. signature of interaction”(p22), refers to sparse results — but it is unclear on first reading which result in particular is ‘sparse’. Also, the structure of the task might not be entirely clear in the reader’s mind, so it could be worth re-stating directly. Also, the next sentence then begins, ‘First...’ but there is not ‘Second...’ to match up with it later on.

Fixed, now near p. 25.

*On p7, an 8th-order Butterworth filter is described with a low-frequency cut-off of 0.05 Nyquist. Please explicitly state this frequency and how it is arrived at (e.g., I’m guessing something like $30 \text{ fps} * 0.5 * 0.05 = 0.75 \text{ Hz}$, or a period of 1.33s?)*

We appreciate the request for more important detail here, and have improved our description of the Butterworth filter. We were guided by parameters for filtering in prior work using frame differencing, admittedly done by inspection of best match to observed raw data, to accommodate greater apparent noise in this video corpus. We acknowledge this more thoroughly in this section and also supply the values requested here (now near p. 8).

On p9, the phrase ‘move in the same way’ is used. However, from my understanding, the way in which people move (suggesting some spatial coordination) is irrelevant — the measure is primarily on the synchrony (and amount) of movement?

Thank you, this has been fixed.

Table 1 is missing information on sex — why?

Sex was a dichotomous code, but we’ve added it now for completeness, including with mean and standard deviation. We modified the name column to indicate that it is dichotomous (Table 1, p. 13), consistent with the PD covariate.

Figure 5 might be better explained - re-state what the axes show in the caption. It could be me, but I need some more information on the jittering process within the main body of the text. Also green and red are not ideal differentiators (a common confusion for colour blind people).

These remarks are very helpful, in looking back we agree that an improvement on this figure and result is called for. We have now completely revised Figure 5, separating those PD outcomes as two panels, and modified our caption (now near p. 22).

Reviewer: 2

The authors aim at testing for the role of behavioral interaction in triadic tasks. The paper is generally well written, and investigates two important topics of interest, namely how joint action coordination behaves differently in larger groups compared to dyads, and what role behavioral synchrony actually serves. The paper is interesting, but I have a couple of methodological issues listed below that the authors need to address.

Thank you for these kind notes and the constructive suggestions. We've addressed each carefully, detailed below.

p.6 It remained unclear which parts of the body were tracked, whether these were different for different groups (or within a group), and which effects this might have on the synchronisation measures.

We have now added several notes of detail about how these regions were chosen (beginning "These regions were delimited in such a way as to...", near p. 7). We appreciate this request as it helps us clarify the data analysis strategy. We also shared concerns about the difference between middle and side interlocutors (beginning "Indeed, an important limitation of the present work is that the triads are captured in the same shot..." near p. 26), as this can affect the measures, as the reviewer notes. We discuss this limitation here, too.

p.7 Please provide the absolute frequency cut-off used, not relative to the Nyquist frequency. Otherwise, it is difficult to judge the reasonability of the cut-off criterion chosen.

We now supply more detail about how we chose these parameters for the Butterworth filter, and include the cutoff range resulting from the low-pass settings (near p. 8, section beginning "In order to find this cutoff, we were guided by prior video differencing research").

p.9 Using cross-correlation because it is „its computation is easy to understand and output relatively transparent“ is probably not the best reason to use this method. The question is, whether it adequately captures the properties of the data, and whether its assumptions are met. Hence, the authors should definitely check for violations of assumptions - linearity, distribution of residuals, presence of outliers - and provide the outcome of the assumption tests in the paper.

We offer an improved justification here. We explicitly aimed to explore how well a common measure of synchrony, time-aligned covariation, mapped onto the variables in our unique data set of interactive outcomes. We have expanded our justification of this by noting that *time-aligned bodily covariation* is the most common measure (abstract, and near p. 9, at section "We tested perhaps the most common measure of synchrony..."). We also add a quite expanded limitations section (as noted below, too) that positions our paper as a kind of illustration that it is important to move to expanded analysis kits (nonlinearity, causality, etc.), and we cite several researchers whose emerging methods are critical for this.

p.12 There is something odd about table 1 for the row „sex“ ...

Sex is a dichotomous code, but we've added it now for completeness, including with mean and standard deviation. We modified the name column to indicate that it is dichotomous (Table 1, p. 13), consistent with the PD covariate.

p.12 I am sorry if I missed this, but how was „triadic correlation“ calculated? p.13 Please describe in more detail, perhaps using an example, how exactly second-order cross-correlation (i.e., the triadic correlation?) was computed. I am not sure whether I correctly understand it, and whether this measure seems to make sense as the label suggests.

We appreciate these requests for more detail. Near p. 11, in section beginning “To assess whether there is *triadic* synchrony...” we supply several new points of detail regarding this “second-order” analysis, as we call it. The extension is meant simply to capture to what extent there is time-alignment in the synchrony of the component dyads. Here’s an example: If two members of the conversation were moving in sync, but a third member as not, then we’d find relatively lower second-order correlation – one dyad would show high synchrony, but the second (with the third member) would depart from this. To get this second-order measure, we correlated windowed correlation time series themselves. In other words, the windowed cross-correlations shown in Fig. 3 (left) are correlated. If the component dyads are covarying similarly, then this second-order correlation should also show a peak at lag 0 higher than a surrogate baseline.

To help readers, we now supply an explicit statement of the second-order correlation (p. 11, section beginning “To assess whether there is *triadic* synchrony”) and have also expanded our figure caption, making it clear that in Fig. 3, left, these time series form the basis of this triadic correlation.

p. 14 Please specify the equation for your MLM. It is hard to tell from your description what was included there and how. For example, if you treat your participant’s data as time series, you might want to include an AR1 residual or similar predictor, but it is not clear whether you did so or not.

Thank you, and we agree: We have now supplied the explicit MLM equations as notes to Tables 2 and 3. As noted above, we also share points about testing for assumptions in our primary inferential tests, and added a check of effects with a simpler parametric test. Additionally, we now share the entire data set (including covariates) and updated scripts on the GitHub repository.

We did not include autoregressive and other relevant covariates for the main reason that our primary *inferential* analyses are not time series in nature. The cross-correlation measures that we extract are themselves used as the raw descriptives for inferential tests (primarily *t*-tests on MLM coefficients). We acknowledge these modeling limitations in the discussion section now, and we also add one-sample *t*-tests on summary statistics as a more standard parametric approach to confirm our results.

We work from a basis of the most common measure of synchrony: measures of *time-aligned covariation* in body signal. So we agree with the reviewer, and include an explicit statement to the effect that, future analyses ought to leverage more complex models (e.g., using Granger logic for flows of influence, see p. 26, section beginning “Expanding measures and improving sensitivity in future analyses may...”).

p. 14 I think it is totally fine to do an exploratory analysis and not correct for multiple testing, but you should still stat the level of significance you are setting.

We now offer a discussion of this cutoff, and more carefully justify our reasoning, near p. 19 in the section of text beginning “To assess potential relationships, we used a conventional significance...”

*In the discussion, I am missing how methodological shortcomings might have resulted in unexpected patterns of effects - null effects and negative effects. It might be, that the time series are not adequately modelled in terms of linear functions (see my point about assumption tests above), or that triadic interactions are not adequately captured by component dyadic interactions (see Wallot, S., Roepstorff, A., & Mønster, D. (2016). Multidimensional Recurrence Quantification Analysis (MdRQA) for the analysis of multidimensional time-series: A software implementation in MATLAB and its application to group-level data in joint action. *Frontiers in psychology*, 7, 1835.), or that the way motion capturing of certain body regions at the expense of others did simply not measure the relevant dimensions of movement... Given the unexpected results and the exploratory nature of the study, these might be other potential sources of what could and could not be observed here.*

We agree with the reviewer and now share much more detailed discussion of these limitations. As noted above, we do use the correlation function as *descriptive measures*, and assess appropriate distributional properties for use of parametric inferential tests (one-sample *t*-test, etc.). However we agree that with varying levels of filtering, the availability of a large ecosystem of potential time series analyses and so on, we would argue that our results generally speak to the importance of moving beyond simple covariation statistics, and we explicitly state this now (pp. 26-28). We cite the suggested paper here, and mention other analyses (e.g., convergent cross mapping, and more).

Appendix B

Dear Drs. Lima and Viding,

Thank you again for another round of helpful remarks. We have again revised our manuscript thanks to the very helpful suggestions from reviewer #1. As before, we include remarks below describing how we made changes to our manuscript. Thank you again for this opportunity to be considered for publication in *Royal Society Open Science*, and we look forward to your response.

Best wishes,

Rick and co-authors

Reviewer 1

Reviewer: 1

This paper is a lovely read, and although the work is largely exploratory, it includes some interesting findings worthy of publication. I thank the authors for their detailed responses to my earlier comments.

Thanks to the reviewer for these very kind comments, and we were happy to address follow-up remarks, mostly through direct changes to our manuscript, as detailed below.

The description of the 2nd-order cross correlation is somewhat clearer now, however I fear it does not go far enough. The new text states that you take the windowed cross-correlations from each of the 3 constituent dyads, you then “calculate a cross correlation” from these. How exactly is this 3-way cross correlation calculated, specifically, how are the 3 cross-correlation graphs combined? (i.e. is there another round of pairwise cross-correlations that then get averaged? Or are the 3 cross-correlations averaged directly?) A diagram would help enormously. It might help to add a specific worked example, perhaps using a fuller expansion of Figure 3 to include the 2nd-order stage?

We appreciate this suggestion, and agree that a new diagram would help. We have now substantially updated that figure, adding much more detail and a specific example (near p. 12).

The value of N , for example, should be made available for every evaluation. A little bit more information might help clear up potential confusion. Specifically, please specify how many dyads are used when calculating the dyadic statistics (e.g. for Tables 1 and 2). If the number of triads is 35, then I assume the number of dyads considered is 3×35 ? However, the text seems to indicate $N=35$. In the PD exploratory study (p21), the number of pairs is given as $N=105$. How is this number arrived at, particularly when there are only 105 participants in total?

We now report this in every test conducted, and also mention that the triad-level factor is integrated in the random effect structure, where relevant. This can be seen on pp. 16, 18, 19, and elsewhere (e.g., see text beginning “specified observed data ($N = 35 \dots$)”).

The repeated measures approach, too, raises some questions. How many 10s windows are used in analysing each dyad/triad? Do these windows overlap (i.e. are they mutually exclusive)? It would also be good to clarify exactly how the windows move. Do they roll forward one raw data sample at a time, or do they jump by a full window-length? Finally, please give some reasoning on why you chose a duration of 10s.

Again this is a great point, and we agree that clarification is needed. We have now clarified the windowed analysis, specifying all these details and why (near p. 10-11, section beginning “The 10-second duration was chosen because...”).

The remainder of my comments concern minor corrections, typos and clarifications.

p6.18 - ‘we predict a variety of individual differences and outcome measures...’ - it would be cleaner if this was more specific, i.e. state how many predictions, followed by a concise list.

We have clarified this now on p. 6 (starting “In order to determine the potential functional role of this synchrony, we...”). Our analysis was simply a single regression model, and this passage incorrectly made it seem as if we were building multiple models (we built just one model that predicts body synchrony from the individual measures and covariates).

p6.29 - ‘This includes _a_ detailed summary...’

Fixed. **And in general:** thank you for these wonderfully detailed suggestions to help with clarification!

p8.46 - ‘...motion signal_s_from similar...’

Fixed.

p8.46 - ‘...0.2 _Hz_ cutoff...’ (missing units)

Fixed.

p8.55 - ‘...before analyzing _the_ main results...’

Fixed.

p10.36 - Briefly justify why 10s. Also, is this a rolling or jumping (non-overlapping) window?

As noted above in this letter, we now add these details and summarize our reasoning.

p14.Table 1 - N value(s)?

As noted above in this letter as well, we now report the *Ns* for each model we run, mainly in the main text around each report.

p16.27-39 - '... a few predicted outcomes...'. This relates to my first comment on the introduction: please enumerate and be more specific about these outcomes.

Fixed also, as noted above.

*p17.18-24 - 'Dyads showed sig... $p < .0001$ ' this sentence is unclear. It opens talking about dyads, but then introduces a one-sampled *t*-test on the triads. Please unpick and clarify.*

Fixed. The statistics, ultimately, are based on component dyads, but it is true that component dyads were aggregated at the triadic level before analysis. We now simply say "triad," as this matches the DF, too. Thank you.

*p17.Table 2 - *N* value(s)?*

As noted above, we now report number of observations for each model in the main text, near each Table.

p18.Figure 4 - middle figure is wrong: the y-axis states 'number of dyads', yet the distribution sums up to 100 (i'm assuming $N=35$, and the distribution is percentages? Please clarify.) A legend indicating observed/surrogate would be helpful on these plots.

We've repaired the figure caption now, clarifying that 105 = the number of dyads (and so this histogram is a reflection of where maximum correlation is lagged for each pair under analysis, near p. 19).

p19.3 - ' $r = .15$ ' ... then later ' $r = .16$ ' ... a little inconsistent

Good catch thank you, we've fixed this (it's $r = 0.16$).

*p20.Table 3 - it might be helpful to specify 'unstandardized beta (*B*)', 'standard error (*SE*)', etc. either in the table or in the text.*

Great point, we've now clarified that it is an unstandardized coefficient (p. 20).

p21.47 - $N=105$ pairs... yet earlier $N=105$ participants. Clarify.

Fixed.